# Histological Changes in the Popliteal Artery Wall in Patients with Critical Limb Ischemia

**DOI:** 10.3390/diagnostics14100989

**Published:** 2024-05-08

**Authors:** Octavian Andercou, Maria Cristina Andrei, Dan Gheban, Dorin Marian, Horațiu F. Coman, Valentin Aron Oprea, Florin Vasile Mihaileanu, Razvan Ciocan, Beatrix Cucuruz, Bogdan Stancu

**Affiliations:** 1Department of Surgery, Emergency County Hospital Cluj, University of Medicine and Pharmacy Iuliu Hatieganu, 400347 Cluj Napoca, Romania; ms26rfl@yahoo.com (F.V.M.); razvan.ciocan@yahoo.com (R.C.); bstancu7@yahoo.com (B.S.); 2Vascular Surgery Department, Satu Mare County Emergency Hospital, 440192 Satu Mare, Romania; mariacristinaandrei@yahoo.com; 3Department of Pathological Anatomy, University of Medicine and Pharmacy Iuliu Hatieganu, 400347 Cluj-Napoca, Romania; dan.gheban@umfcluj.ro; 4Second Surgical Department, Emergency County Hospital Mures, University of Medicine and Pharmacy Emil Palade, 540142 Targu Mures, Romania; dorinmarms@yahoo.com; 5Department of Vascular Surgery, Emergency County Hospital, 400006 Cluj Napoca, Romania; horatiucoman@yahoo.com; 6Department of Surgery, Emergency Military Hospital Cluj Napoca, University of Medicine and Pharmacy Iuliu Hatieganu, 400347 Cluj-Napoca, Romania; opreacv31@gmail.com; 7Department of Vascular Surgery, Martha Maria Hospital Nuremberg, 90491 Nuremberg, Germany; beatrix.cucuruz@martha-maria.de

**Keywords:** diabetes mellitus, atherosclerosis, medial calcinosis, atherosclerotic plaque critical limb ischemia

## Abstract

Introduction: This prospective study aims to illustrate the histopathological arterial changes in the popliteal artery in peripheral arterial disease of the lower limbs. Material and method: A total of 60 popliteal artery segments taken from patients who had undergone lower limb amputation were examined between April and June 2023. The degree of arterial stenosis, medial calcinosis, and the vasa vasorum changes in the arterial adventitia were quantified. The presence of risk factors for atherosclerosis was also observed. Results: Atherosclerotic plaque was found in all of the examined segments. Medial calcinosis was observed in 40 (66.6%) of the arterial segments. A positive association between the degree of arterial stenosis and the vasa vasorum changes in the arterial adventitia was also found (*p* = 0.025). The level of blood sugar and cholesterol were predictive factors for the severity of atherosclerosis. Conclusions: Atherosclerosis and medial calcinosis are significant in patients who underwent lower limb amputation. Medial calcinosis causes damage to the arterial wall and leads to a reduction in responsiveness to dilator stimuli.

## 1. Introduction

Peripheral arterial disease (PAD) encompasses a spectrum of vascular disorders characterized by the gradual occlusion of arteries, predominantly affecting the lower extremities. Critical limb ischemia (CLI), an advanced and severe manifestation of PAD, is characterized by insufficient blood flow to the extremities, resulting in rest pain, chronic ulcers, and, in severe cases, gangrene. CLI poses a significant threat to limb viability and is associated with an increased risk of major amputations, significantly impacting patients’ quality of life [1,2]. 

Despite advances in endovascular and surgical interventions, delayed diagnosis and suboptimal management contribute to the persistent occurrence of major amputations in patients with CLI. The amputation rate in the general population suffering from PAD is on the decrease; however, amputations continue to be performed despite the development of advanced revascularization techniques The lack of precise criteria defining salvageable limbs exacerbates this challenge, underscoring the need for a deeper understanding of the histopathological changes within the arterial walls of affected limbs [3,4].

Atherosclerosis, characterized by the formation of atherosclerotic plaques within arterial walls, is a predominant contributor to PAD and CLI. These plaques lead to luminal stenosis, compromising blood flow to the lower extremities [5]. Additionally, medial calcinosis, the deposition of calcium in the media of blood vessels, is a common alteration observed in patients with diabetes mellitus associated with peripheral arterial diseases. Even though little is known about the etiology or exact significance of this pathology, medial calcinosis was described as part of other diseases, such as diabetes mellitus and the last stage of chronic kidney disease [6,7,8,9].

The proposed study seeks to address these knowledge gaps by conducting an in-depth histopathological analysis of arterial segments obtained from patients who have undergone lower limb amputation due to CLI. The investigation will encompass a detailed examination of arterial changes, including the degree of stenosis, morphopathological features, and classifications of atherosclerotic lesions. Furthermore, this study aims to explore the association between these arterial changes and established risk factors for atherosclerosis, such as diabetes mellitus and dyslipidemia [10,11]. It is worth noting that individuals with diabetes mellitus and dyslipidemia generally run a high risk of developing peripheral atherosclerotic disease [12,13].

Understanding the intricacies of arterial changes in CLI patients holds potential clinical significance. Insights into the severity and patterns of atherosclerosis, coupled with an examination of medial calcinosis, can guide clinicians in refining treatment strategies, making informed decisions regarding the level of amputation, and potentially improving outcomes for individuals with CLI [14,15]. Additionally, the study seeks to establish correlations between arterial changes and specific risk factors, paving the way for more targeted interventions and personalized management approaches [16].

By highlighting the histopathological changes in the arterial wall in CLI, this research aspires to contribute valuable knowledge that can enhance clinical decision making, enhance risk stratification, and ultimately lead to improved care for patients struggling with the debilitating consequences of critical limb ischemia.

## 2. Materials and Methods

The institutional review board at our institution approved the protocol for this study under the number 12669/21 March 2023.

This was a transversal study that included 60 consecutive patients hospitalized in the Surgical Department of the Emergency University Hospital, Cluj-Napoca, Romania, from April to June 2023. The present study included patients who had first-intention above-knee amputation for severe ischemic lesions. Patients with amputations due to other causes (i.e., after failed revascularizations) and patients with irreversible tissue changes after acute ischemia were excluded. Following above-knee amputation, limb dissection occurred within 10 min to 1 h. Standardized techniques were employed to ensure consistency in the dissection process. The dissection aimed to expose the third (distal) segment of the popliteal artery. A total of 3 cm of popliteal artery was collected proximally, from above its bifurcation. The pieces thus obtained were placed in 10 percent formaldehyde solution. Subsequently, sections of the pieces were cut, which were then sent for fixing in paraffin. All sections were examined under an optical microscope in hematoxylin-eosin staining. The samples were examined using a Leica (Danaher Corporation, Washington, DC, USA) DM750 optical microscope with a Leica ICC50HD image acquisition system. A qualified pathologist, with expertise in vascular pathology, conducted the examination.

The following parameters were monitored: the degrees of lumen stenosis (first degree = 0–50%; second degree = 50–75%; third degree = over 75% of the lumen is obstructed). Lesions were classified according to Stary et al. criteria [17]. Criteria for the layout of the atherosclerotic plaque were based on type 4 (atheroma) = extracellular lipid deposits in the shape of cholesterol crystals; type 5 = fibroatheroma: 5a—fibro lipid plaque, 5b—calcified plaque, 5c—fibrous plaque; type 6 = complicated plaque with ulceration, fissure, thrombosis, hemorrhage, or intense calcification); the layout of the atherosclerotic plaque (circumferential, localized). The presence or absence of medial calcinosis was documented as localized or circumferential. (absent; present—localized or circumferential).

Data on risk factors (diabetes mellitus, hypertension, hypercholesterolemia, smoking) were collected using standardized forms. Cut-off values for risk factors were defined as per the American Diabetes Association and other relevant guidelines [10,11]: blood sugar level greater than 120 mg/dl, hypertension (systolic blood pressure greater than or equal to 130 mmHg or diastolic blood pressure greater or equal to 80 mmHg), hypercholesterolemia (total cholesterol value greater than 200 mg/dl), and smoking.

Statistical analysis was performed using SPSS medical statistics software (version 13.0) and Microsoft Excel. Statistical data processing was carried out using the Shapiro–Wilk test, and, based on the results of this test, it was decided which of the following tests to apply: the Mann–Whitney U test, ANOVA quantitative univariate analysis, or, to be more precise, the multiple comparison post hoc analysis (the Bonferroni test), or the Chi-Square test, in which we monitored the value of *p* with an alpha significance threshold set to 0.05.

Informed consent was obtained from patients or their legal representatives before inclusion in the study. Measures were implemented to ensure patient confidentiality and compliance with data protection regulations.

## 3. Results

In this study, we examined 60 arterial segments from the amputated lower limbs. The clinical conditions which led to amputation included a wide range of situations. Of the subjects included, 32 (53.3%) had chronic ulceration without any healing tendency in the calcaneus area, while 28 (46.6%) showed moist or dry gangrenous lesions in the hallux or forefoot area. Other local manifestations included edema, rash, and cellulitis. All patients experienced rest pain in the affected limb. All patients have been investigated by CT angiography or digital subtraction angiography and confirmed the absence of indications for lower limb revascularization, with amputation being the sole surgical indication.

The demographic and clinical data and the risk factors for the peripheral atherosclerotic disease are shown in Table 1.

Each patient was documented for coronary artery disease, cerebrovascular disease, and chronic kidney disease. Of the 60 patients, 59% (*n* = 35) had coronary artery disease, 17% (*n* = 10) had cerebrovascular disease, and 45% (*n* = 27) had documented chronic renal disease.

Patients diagnosed with high cholesterol levels were on treatment with statine at the time of admission to the hospital, but we did not look for correlations between the time of treatment and arterial wall modifications. 

The histomorphology analysis of the arterial segments reveals atherosclerotic plaques in all of the examined cases. Three (5%) of the examined popliteal artery segments showed extracellular lipid deposits in the shape of cholesterol crystals (Figure 1). In total, 6 (10%) of the examined popliteal artery segments had fibro lipid plaques (Figure 2); 25 (41.6%) had calcified plaques (Figure 3); and in 14 (23.3%) of the examined segments, the intima was thickened and replaced by fibrotic connective tissue, and the quantity of lipids was minimal (Figure 4). Twelve (20%) of the segments presented complicated atherosclerotic plaques.

Of the intimal atherosclerotic plaques, 35 (58.3%) had a circumferential layout, while 25 (41.6%) had a focal (localized or graded) layout in the arterial wall (Figure 5).

First-degree stenosis (less than 50%) was seen in 25% of the segments; second-degree stenosis (50–75%) was seen in 26.6% of the segments, and third-degree stenosis (over 75%) was seen in 48.3% of the segments. 

The vasa vasorum in the adventitia were narrowed or even totally or partially blocked through endothelial proliferations in 56 (93.3%) of the examined arterial segments, and only 4 (6.6%) of the examined arterial segments did not show any changes in the vasa vasorum of the adventitia. A positive association was also found between the degree of arterial stenosis and the vasa vasorum changes in the arterial adventitia (Chi-Square test χ^2^ = 11.762, *p* = 0.003; Kendall’s tau-b = 0.370, *p* = 0.025). 

Medial calcinosis was absent in 20 (33.3%) of the examined segments. The media were invaded by an atheromatous process in 40 (66.6%) of the segments, the muscle fibers degenerated and disappeared, and calcium deposits appeared, which sometimes determine bone hardness over large artery portions.

Statistically significant differences were described between the average cholesterol level and the type of atherosclerotic lesion of the popliteal artery (*p* = 0.001). Statistically significant differences were found between the type of atherosclerotic lesion and the average blood sugar level (*p* = 0.003). 

Significant differences were found between the average cholesterol values and the medial calcinosis of the popliteal artery (*p* = 0.001); the average cholesterol values in the absence of medial calcinosis or in the presence of nodular medial calcinosis are significantly lower than the average cholesterol values in circumferential medial calcinosis (*p* = 0.018 vs. *p* = 0.001). Significant differences were also found between the average blood sugar values and the medial calcinosis of the popliteal artery (*p* = 0.012); the average blood sugar values in nodular medial calcinosis are significantly lower than in the case of circumferential medial calcinosis (*p* = 0.001). 

These findings offer a detailed insight into the histopathological characteristics of arterial segments, highlighting significant associations between atherosclerotic lesions, vasa vasorum changes, and key risk factors and contributing to a better understanding of the vascular pathology in this specific patient cohort.

## 4. Discussion

Atherosclerotic peripheral vascular disease, when diagnosed late in the course of the disease, often leads to significant changes in a patient’s comfort, as well as disabilities, which eventually lead to the amputation of the affected limb. Amputation, which, through surgery, leads to bodily deformities, is an accepted method for the improvement of an individual’s quality of life, for relieving pain in the nonfunctional limb, and for the removal of a persistent and recurrent infection focus. It is a rather common procedure in individuals with extensive tissue lesions without any healing tendency and without any possibility for vascular reconstruction. 

The average age of the patients was 67, and most of the patients were males (73.3%), which is in accordance with the data provided in the literature [2,18].

The popliteal artery was examined near the bifurcation, because it is known that atherosclerotic lesions tend to be more severe in bifurcation areas also due to the hemodynamic changes in these very areas.

The histological classification of atherosclerotic lesions demonstrates the severity of the atherosclerotic disease in the patients included in our study. We used the advanced histological classification of atherosclerotic lesions developed by Stary in 1995 [17] Our study, focusing on histopathological arterial changes, can benefit from recent advancements in noninvasive carotid wall imaging [19]. While our primary emphasis is on lower limb arteries, the integration of noninvasive imaging findings broadens our understanding of systemic atherosclerosis. This inclusion aligns with the growing recognition of the interconnected nature of vascular diseases, emphasizing the need for a holistic approach to arterial assessment.

It is a known fact that high cholesterol and blood sugar levels are associated with atherothrombosis risk. In our study, we found that an advanced stage of atherosclerotic lesions is correlated with high blood cholesterol and high blood sugar levels, a fact attested by the data in the literature [20]. As discussed by Belur et al. [21], the role of lipid-lowering therapy in peripheral artery disease is crucial. Our study’s findings on the severity of atherosclerosis correlate with this understanding. The integration of lipid management strategies becomes paramount in the context of critical limb ischemia, potentially influencing the progression of arterial changes and patient outcomes. This study prompts a reflection on the potential impact of lipid-lowering therapies on mitigating atherosclerotic lesions in our patient population.

More than three decades ago, the distribution of the vasa vasorum heterogeneity in the vascular wall was demonstrated, but it was not correlated with the regional tendency for the formation of atheroma plaques [22].

In our study, almost all examined artery segments showed changes in the vasa vasorum of the adventitia: the higher the degree of lumen stenosis in the artery segments, the more severe these changes were. In the literature, there are studies that attest to the fact that vasa vasorum play a part in the initiation, progress, and complications of atherosclerosis [23].

Frese et al.’s proposed diagnostic scoring system for distinguishing chronic critical limb ischemia from intermittent claudication adds a valuable dimension to our study [24]. While our primary focus is histopathological, the integration of a diagnostic scoring approach aligns with our goal of assessing the severity of arterial changes. This scoring system offers a potential tool for clinicians to refine their diagnostic and treatment decisions, providing a more nuanced understanding of critical limb ischemia.

Medial calcinosis was described for the first time by Mönckeberg in 1903, as a progressive condition discovered in peripheral arteries in elderly but apparently healthy patients. Medial calcinosis and atherosclerosis may be found in the same vessels; they may manifest independently or differently: medial calcinosis is associated with smooth and elastic muscle fiber dysfunction, while atherosclerosis is the consequence of endothelial dysfunction. Concretely, medial calcinosis consists of a calcium deposit in the blood vessel media. A linear calcium deposit, which is initially seen in the internal elastic lamina, progresses circumferentially, the calcium crystals being surrounded by smooth muscle fibers. Because this calcium accumulation does not obstruct the blood flow, medial calcinosis is often considered to be an insignificant finding, but medial calcinosis reduces the vessel’s elasticity and distensibility, as well as its capacity to respond to vasodilation and vasoconstriction stimuli, which may increase the development of intimal lesions, such as atherosclerosis [25].

The study by Santillán-Cortez et al. investigates the correlation between endothelial progenitor cells (EPCs) and major amputation after angioplasty in critical limb ischemia patients [26]. Considering this cellular aspect enriches our understanding of the disease process. The potential link between EPCs and major amputation raises intriguing questions about the role of biomarkers in predicting patient outcomes and guiding interventions. Integrating biomarker exploration into our discussion opens avenues for a more personalized and targeted approach to patient care.

This calcification develops in association with comorbidities such as a high level of blood sugar and blood cholesterol [27]. 

An association between blood sugar level and the type of medial calcinosis layout has shown that the blood sugar level in patients with circumferential medial calcinosis is greater than in the case of patients with nodular medial calcinosis; however, these differences are questionable because of the small number of patients with circumferential medial calcinosis, and, in order to validate these differences, this study needs to be extended so as to comprise a larger number of patients. A correlation between the severity of medial calcinosis and cholesterol level was also found in our study. A low cholesterol level was found in patients without medial calcinosis of the popliteal artery, as compared with those with medial calcinosis. 

Tigkiropoulos et al. introduce a polymer-free Amphilimus drug-eluting stent for infrapopliteal arterial disease [28]. While not directly aligned with our histopathological focus, this innovation prompts consideration in our discussion of vascular revascularization approaches. The introduction of advanced stenting devices raises questions about their potential efficacy in our patient population, especially in cases where traditional revascularization techniques may be limited.

### 4.1. Limitations of this Study

While this study provides valuable insights into popliteal artery wall changes in patients with critical limb ischemia, it is important to acknowledge certain limitations that may impact the interpretation and generalization of the findings. This study included a relatively small sample size of 60 arterial segments, limiting the generalizability of the results to a broader population. The findings may not be representative of the entire population with critical limb ischemia. Although we had a significant statistical number of patients, this study was conducted at a single institution, which may introduce biases related to the characteristics of the patient population, local practices, and available resources. Multicenter studies are recommended for more diverse and representative results. Another limitation of our study is that the inclusion criteria, such as patients undergoing first-intention above-knee amputation for severe ischemic lesions, may introduce selection bias. While this study considered risk factors such as diabetes mellitus, hypertension, hypercholesterolemia, and smoking, it is important to note that other potentially confounding variables, such as genetic factors or specific medication use, were not thoroughly addressed; but, in our study, we decided to investigate only the most frequent risk factors associated with critical limb ischemia. Addressing these limitations in future research could contribute to a more robust understanding of arterial changes in critical limb ischemia.

### 4.2. Clinical Implications

Despite the limitations of this study, we consider that the clinical implications of the arterial wall changes in patients with critical limb ischemia are significant and can influence various aspects of patient care, treatment strategies, and preventive measures. Here are some key clinical implications.

This study highlights the predictive role of blood sugar and cholesterol levels in the severity of atherosclerosis. Clinicians should prioritize tight control and management of blood sugar and cholesterol levels in patients with critical limb ischemia to potentially slow the progression of arterial changes.

Since PAD, if not treated promptly, can lead to major amputations, our study emphasizes the importance of early detection and intervention. Clinicians should be vigilant in identifying patients with critical limb ischemia, especially those with diabetes mellitus and dyslipidemia who are at a higher risk.

The information provided by our study can aid surgeons in making decisions about the level of amputation, considering the severity and extent of atherosclerosis and medial calcinosis.

Understanding the histological characteristics of arterial lesions can be valuable in planning revascularization procedures. This study suggests that advanced peripheral atherosclerotic disease may limit the effectiveness of revascularization techniques, emphasizing the need for alternative approaches or early interventions such as endovascular procedures.

Given the association between risk factors (e.g., diabetes, high cholesterol) and arterial changes, this study underscores the importance of patient education. Encouraging lifestyle modifications, such as diet and exercise, becomes crucial to mitigate risk factors and prevent the progression of atherosclerosis.

Patients with critical limb ischemia may benefit from regular surveillance to monitor the progression of arterial changes. This can help in adjusting treatment plans and interventions based on the evolving pathology.

This study lays the groundwork for future research investigating the relationship between specific risk factors, arterial changes, and outcomes. This can lead to the development of targeted interventions and therapies to address the underlying pathophysiology.

The findings of this study, once validated through further research, may contribute to the development or modification of clinical guidelines for managing patients with critical limb ischemia. This could include recommendations for regular screening, risk factor management, and personalized treatment approaches.

## 5. Conclusions

In our study, histopathological arterial lesions in amputated limbs were found in all of the examined vessels. The blood sugar and blood cholesterol levels were predictive factors for the severity of atherosclerosis. The patients who had circumferential medial calcinosis had a higher blood sugar level. This study has several clinical implications that extend to risk factor management, surgical decision making, patient education, and potential advancements in treatment strategies for patients with critical limb ischemia. Implementing the insights gained from this research into clinical practice has the potential to improve patient outcomes and guide healthcare professionals in providing more targeted and effective care.

## Figures and Tables

**Figure 1 diagnostics-14-00989-f001:**
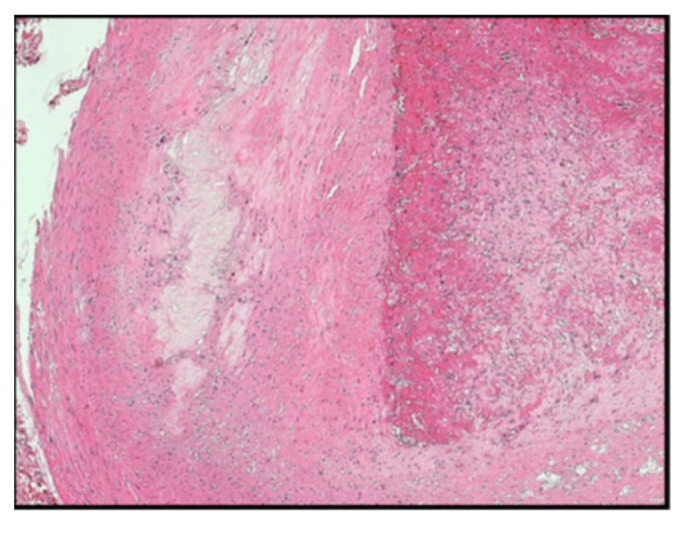
The atheroma stage (type 4 according to Stary’s classification), with extracellular lipid deposits in the shape of cholesterol crystals (hematoxylin-eosin staining).

**Figure 2 diagnostics-14-00989-f002:**
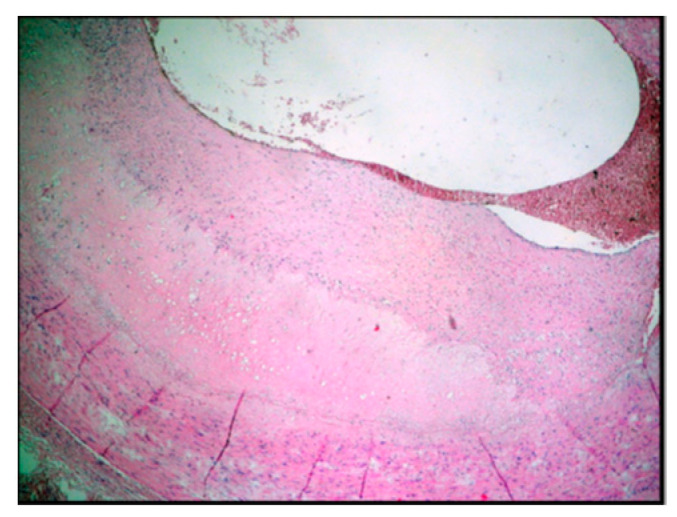
Fibro lipid plaque (type 5 according to Stary’s classification).

**Figure 3 diagnostics-14-00989-f003:**
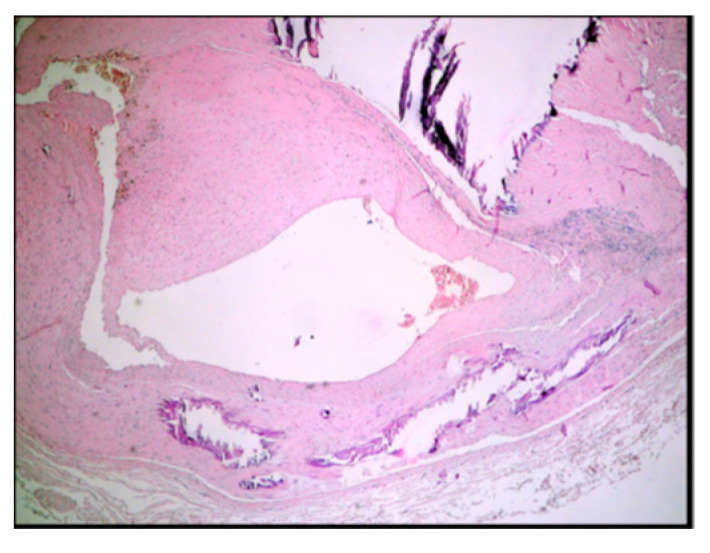
Calcific lesions in the atheroma plaque (type 5B according to Stary’s classification).

**Figure 4 diagnostics-14-00989-f004:**
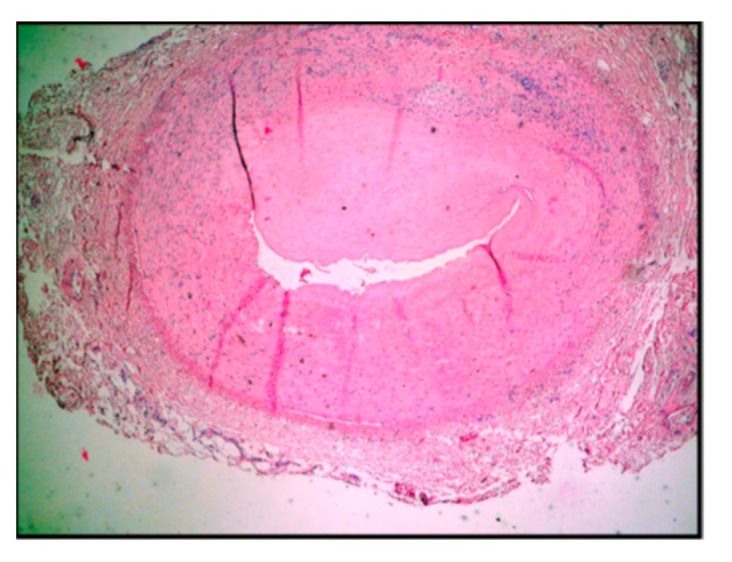
Fibrose plaque, in which the intima is thickened and replaced by fibrotic connective tissue, and the lipids are in minimal quantity or even absent (type 5C according to Stary’s classification).

**Figure 5 diagnostics-14-00989-f005:**
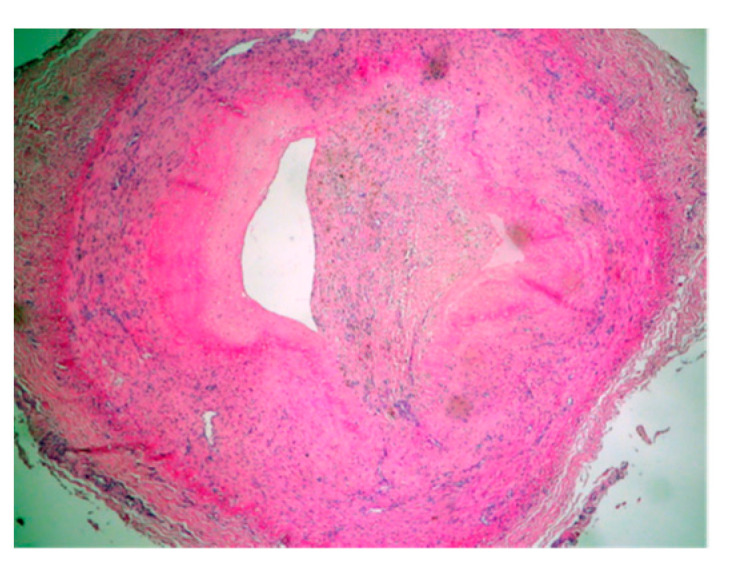
Complicated atheroma plaque with ulceration, fissure, thrombosis, and hemorrhage (type 6 according to Stary’s classification).

**Table 1 diagnostics-14-00989-t001:** Patient’s data.

Characteristics	Patients (*n* = 60)
Age (years)	67.87 ± 8.625
Sex (male/female)	42 (73.3%)/18 (26.7%)
Smoking	32 (70%)
Systolic Blood Pressure levels	156.53 ± 25.428; 95%CI
Blood glucose level	142.43 ± 75.671; 95%CI
Blood cholesterol level	185.24 ± 42.828; 95%CI

## Data Availability

Data is contained within the article.

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
