# Peer review of "Histological Changes in the Popliteal Artery Wall in Patients with Critical Limb Ischemia"

_diagnostics, 2024, doi:10.3390/diagnostics14100989_

Round 1

Reviewer 1 Report

Comments and Suggestions for Authors

The authors present an original study entitled “Histological Changes of the Popliteal Artery Wall in Patients with Critical Limb Ischemia”.

The aim of the study was to demonstrate the histological features of the popliteal artery wall in patients with PAD undergoing lower limb amputation.

The results of the study are interesting, however, I have a few points to address:

1.    The authors describe the exclusion criteria: “patients with irreversible tissue changes after acute ischemia, were excluded”. Further, providing a clinical overview of the patients, the authors state: “while 28 (46.6%) showed moist or dry gangrenous lesions in the hallux or forefoot area”. Obviously, these were also patients with irreversible tissue changes. Please clarify this point when describing the exclusion criteria.

2.    “Lesions were classified according to Stary et al criteria”. Please provide, references.

3.    Сut-off values for risk factors were defined as per American Diabetes Association and other relevant guidelines”. Please provide, references.

4.    In Table 1, the authors provide data in the form of mean and standard deviation. The 95% confidence interval is then additionally given. Please provide data in the format of mean and standard deviation (for values with normal distribution) or leave only the median and 95% confidence interval.

5.    A typo in Table 1 – “156.53±25.428; 95%CI (147.04; 16.03)”.

6.    Table 1 shows BP, glucose and cholesterol levels. However, no information is provided about the medical therapy received.  In the following, the authors also analyse the relationships and differences in cholesterol and glucose levels according to histological features. However, it is not clear whether these patients were naive to lipid-lowering therapy?

7.    The Discussion section is very extensive and needs to be reduced.

Author Response

Thank you for the time to review our paper. Attached please find the responses to your comments.

With best regards,

Octavian Andercou

Reviewer 2 Report

Comments and Suggestions for Authors

The manuscript is of potential interest, although the novelty is relatively low. Authors should provide more detailed description of the study population and review English for limiting typos and errors. Some specific comments are reported below: 

On page 2, lines 77-78, please clarify in which city/state the study has been conducted. 

On page 3, line 101, please change "high blood pressure" to "hypertension". Authors should also provide diagnostic criteria for each fo the cardiovascular risk factors listed here and considered for the study purposes.

On page 3, line 118, please provide data on how many patients/segments have been excluded and main reasons for being excluded.  

On table 1, authors should provide more data on diastolic blood pressure "levels" (not "level"), BMI, serum creatinine and cardiovascular risk factors and comorbidities.

Figures are too large and should be reduced, possibly to be included in a single box of images. 

Some references appear to be different from other ones. 

Comments on the Quality of English Language

English editing should be applied to eliminate typos

Author Response

(The authors gave the same response as above.)
